# Carotid Artery Stenting in Patients with Atrial Fibrillation: Direct Oral Anticoagulants, Brief Double Antiplatelets, and Testing Strategy

**DOI:** 10.3390/jcm10225242

**Published:** 2021-11-11

**Authors:** José E. Cohen, John Moshe Gomori, Asaf Honig, Ronen R. Leker

**Affiliations:** 1Departments of Neurosurgery and Radiology, Hadassah-Hebrew University Medical Center, Jerusalem 91120, Israel; 2Department of Radiology, Hadassah-Hebrew University Medical Center, Jerusalem 91120, Israel; gomori@md.huji.ac.il; 3Department of Neurology, Hadassah-Hebrew University Medical Center, Jerusalem 91120, Israel; asaf.honig2@gmail.com (A.H.); leker@hadassah.org.il (R.R.L.)

**Keywords:** anticoagulation, antiplatet regimen, atrial fibrillation, carotid artery stenting, carotid stenosis, ischemic stroke

## Abstract

Carotid endarterectomy is usually preferred over carotid artery stenting (CAS) for patients with atrial fibrillation (AF). We present our experience with short-course periprocedural triple antithrombotic therapy in 32 patients aged >18 years with nonvalvular AF undergoing CAS. There were no deaths, cardiac events, embolic strokes, hyperperfusion syndrome, intracranial hemorrhage, or stent thrombosis within 30 days. Transient intraprocedural hemodynamic instability in 15/32 (47%) and prolonged instability in 4/32 (13%) was managed conservatively. At a mean 16-month follow-up, there were no new neurological events or deterioration. Mean stenosis was reduced from 78.0% ± 9.7% to 17.3% ± 12.2%. This retrospective study included patients AF who were symptomatic (minor stroke (NIHSS ≤ 5)/TIA) with ICA stenosis >50%, or asymptomatic under DOAC therapy with carotid stenosis >80%, who underwent CAS from 6/2014–10/2020. Patients received double antiplatelets and statins. Antiplatelet therapy effectiveness was monitored. Stenting was performed when P2Y12 reaction units (PRU) were <150. DOACs were discontinued 48 h before angioplasty; one 60 mg dose of subcutaneous enoxaparin was administered in lieu. DOAC was restarted 12–24 h after intervention. Patients were discharged under DOAC and one nonaspirin antiplatelet. 32 patients on DOAC were included (26 male, mean age 71). 19 (59.4%) presented with stroke (ICA stenosis-related in 14); 13 (40.6%) were asymptomatic. Stents were deployed under filter protection following pre-angioplasty; post-angioplasty was performed at least once in 12 patients (37.5%). Our experience suggests that CAS can be safely performed in selected patients with CAS and AF requiring DOAC. The role of CAS in AF patients under DOAC warrants study in rigorous trials.

## 1. Introduction

Carotid artery stenting (CAS) is usually withheld in favor of carotid endarterectomy for patients with atrial fibrillation (AF), primarily because of the need for double antiplatelet therapy to prevent stent-related thromboembolic complications, and anticoagulation with direct oral anticoagulants (DOAC) to prevent cardioembolic stroke during stenting. On the other hand, carotid endarterectomy is associated with increased complication rates in patients receiving perioperative antithrombotic therapy [1,2].

Data on triple antithrombotic therapy after carotid stenting is lacking, but this strategy has been studied extensively by interventional cardiologists [3,4,5], since a significant proportion of patients with AF undergo percutaneous coronary interventions (PCI) [5]. When these patients undergo PCI or have an acute coronary syndrome, DOAC and double antiplatelet therapy (DAT), usually with aspirin and clopidogrel, have been combined. However, this triple antithrombotic strategy has been scrutinized because it increases the risk of serious bleeding [4]. Modified regimens that reduce bleeding risk without increasing the incidence of coronary or cardioembolic events have been explored. In the absence of guidelines, these regimens are determined for individual patients based on case-by-case assessment of three competing risks: cardioembolic stroke, coronary ischemic events, and bleeding [5].

A recent systematic review and meta-analysis [3] of randomized controlled trials focused on the effect of aspirin-omitted double antithrombotic therapy (AODAT) on coronary ischemic events and compared AODAT with triple agent therapy in nonvalvular AF patients presenting with acute coronary syndrome or undergoing percutaneous coronary interventions. The analysis concluded that AODAT reduces the occurrence of bleeding episodes, but found a higher rate of myocardial infarction and stent thrombosis in nonvalvular AF patients presenting with acute coronary syndrome or undergoing percutaneous coronary interventions [6]. Another group [7] that recently investigated the efficacy and safety of different antithrombotic strategies in patients with AF undergoing interventions concluded that for those undergoing coronary interventions, treatment with apixaban + P2Y12 inhibitors was associated with lowest bleeding rate compared with other regimens, including other DOACs + P2Y12 inhibitors. There was also no increase in ischemic outcomes.

CAS requires intraprocedural antiplatelet therapy to reduce the incidence of acute thromboembolic complications and stent thrombosis. From the published research on coronary interventions, we know that most thrombotic events occur within hours of the acute stenting procedure; therefore, antithrombotic drugs must be administered during the intervention [8]. Guidelines on carotid endarterectomy and stenting are unanimous in advising perioperative continuation of antiplatelet therapy for all patients to prevent thromboembolic events; however, there is no specification of the type of antiplatelet therapy [9] and recommendations on DAT are inconsistent.

The encouraging experience in AF patients who underwent percutaneous coronary interventions and were successfully managed with DOACs and single P2Y12 inhibitors prompted us to explore a strategy consisting of a brief period of periprocedural triple antithrombotic therapy followed by DOACs + P2Y12 inhibitors in patients with AF undergoing CAS.

## 2. Materials and Methods

### 2.1. Patients

This retrospective study included patients with nonvalvular AF under novel oral anticoagulants who underwent CAS for the management of carotid artery stenosis at our center between June 2014 and October 2020. The requirement for informed consent was waived by our Institutional Review Board (0405-20-HMO).

Stenosis was initially diagnosed by CT angiography (CTA) in symptomatic patients and by either Doppler sonography and/or CTA in asymptomatic patients. All patients underwent diagnostic digital subtraction angiography (DSA). The degree of carotid stenosis was determined based on North American Symptomatic Carotid Endarterectomy Trial (NASCET) criteria [10].

Inclusion criteria were:Age > 18 yearsPatients who were symptomatic after minor stroke (National Institutes of Health Stroke Score [NIHSS] ≤5 or transient ischemic attack (TIA, cerebral or retinal) and who presented atherosclerotic internal carotid artery (ICA) stenosis >50%, orAsymptomatic patients under DOAC therapy for nonvalvular AF who had severe carotid stenosis over 80% or confirmed progression of stenosis

To estimate the stroke risk in these patients we used the CHA2DS2 VASc score [11]. CHA2DS2 stands for Congestive heart failure, Hypertension, Age, Diabetes, previous Stroke/transient ischemic attack. VASc stands for vascular disease (peripheral arterial disease, previous myocardial infarction, aortic atheroma), and gender (female).

We excluded patients with symptomatic carotid stenoses presenting with moderate or major stroke requiring urgent endovascular revascularization procedures or intracranial thrombectomy, those with tandem stenoses requiring concomitant intracranial stenting, those under warfarin therapy, those with vascular anatomy that was unsuitable for endovascular management, those without adequate percutaneous vascular access, and individuals with major contraindications to antiplatelet therapy.

### 2.2. Preparation for the Angioplasty Procedure

Patients presenting with TIA or minor stroke were treated during the first 2 weeks after the presenting event. In addition, to reduce the incidence of renal function deterioration, patients with renal failure underwent selective DSA (Allura Xper; Philips Healthcare, Best, The Netherlands) to confirm stenosis characteristics and evaluate procedure feasibility using a reduced contrast dose. The angioplasty procedure was routinely performed 1 week after DSA. All patients signed the informed consent form according to our protocol.

Patients were instructed to take aspirin (100 mg per day) and clopidogrel (75 mg per day) for at least 4 days before the procedure. When this was not possible, they received a loading dose of clopidogrel (300 mg) and aspirin (300 mg) on the day of the procedure. In addition, patients were routinely instructed to take atorvastatin 80 mg/day or rosuvastatin 40 mg/d. Patients were kept on their usual medications except for DOACs, which were discontinued 48 h before the angioplasty procedure; a single dose of 60 mg of subcutaneous enoxaparin (Clexane, Sanofi Aventis, Paris, France) was administered 24 h before instead. Antidiabetic medications were discontinued 12 h before the intervention. All patients sustained evaluation of thrombocyte inhibition levels using the VerifyNow P12Y12 assay (Accumetrics, San Diego, CA, USA). Patients were treated only when P2Y12 reaction unit (PRU) levels were less than 150, ideally 60–150, and the antiplatelet regime was modified in accordance. If the response was lower and without resistance, additional loading doses or increased daily doses (for example, 150 mg daily) were administered. If clopidogrel resistance was detected, clopidogrel was discontinued and ticagrelor or prasugrel was prescribed.

All patients underwent pre- and postoperative neurological evaluations by specialized neurovascular physicians. In addition, patients with low cardiovascular risk underwent a baseline electrocardiogram and echocardiogram; those with a high-risk cardiovascular profile were evaluated by senior cardiologists before the endovascular procedure, and had tailored complementary cardiovascular evaluations and medications based on this evaluation.

### 2.3. Angioplasty Procedure

All procedures were performed by a neuroendovascular neurosurgeon (J.E.C.) and neuroradiologist (J.M.G.), both with extensive experience performing neuroendovascular procedures. Cervical and cerebral DSA were performed via a femoral approach using a 4 French diagnostic catheter. Carotid artery access, ICA stenosis characteristics, primary and secondary (leptomeningeal) flow compensation, and any anastomoses between the internal and external carotid arteries were evaluated. If the lesion was considered suitable for stenting, the 4 French introducer sheath was exchanged for a 7–8 French introducer sheath. Procedures were performed under local anesthesia when possible, with or without sedation, or under general anesthesia when indicated. Bilateral stenoses were treated in two different sessions with an interval of at least 1 month between procedures.

Electrocardiogram, invasive blood pressure, and blood oxygen saturation levels were continuously checked, and intravenous fluids, atropine, and dopamine were prepared to respond to intraprocedural hemodynamic instability. Intravenous heparin was then administered to achieve an activated clotting time of 250–300 s. Atropine was not prophylactically administered before stent deployment; it was used only if bradycardia occurred during the procedure.

A 90 cm 7–8 French guiding catheter (Guider Soft Tip, Boston Scientific, Marlborough, MA, USA) was advanced into the distal common carotid artery over an exchange guidewire placed in the external carotid artery. Under road mapping, a 0.014″ guidewire (Transend, Boston Scientific, Marlborough, MA, USA) was used to navigate a filter-type embolic protection device (Spider, eV3/Covidien, Plymouth, MN, USA) through the cervical ICA for deployment distal to the selected landing segment. In the case of tortuous vascular anatomy that precluded safe or secure landing of the filter, balloon predilation was performed. In the event of bradycardia after balloon deflation, atropine was administered. The stent delivery system was then advanced through the stenosis and deployed across the stenosis. The degree of residual stenosis was evaluated and post-dilation was performed only when considered necessary. Cerebral angiogram was obtained at the end of the procedure to rule out procedural emboli. Hemostasis of the puncture site was achieved by manual compression or with a percutaneous closure device (Angio-Seal, St Jude Medical, Minnetonka, MN, USA).

After the procedure, all patients remained in the intensive care unit for 12 h. If they were clinically and hemodynamically stable, they were then transferred to the Neurology or Neurosurgery Department for 36 h. An immediate post-angioplasty neurologic evaluation was performed by a neurovascular specialist and repeated after 12 and 24 h. Dual antiplatelet therapy was continued for one day after the intervention and then aspirin was discontinued. DOAC was usually restarted 12–24 h after the intervention. Patients were regularly discharged under DOAC and a single P2Y12 inhibitor, with doses targeted for compliance with our PRU target. Antiplatelet inhibition was reevaluated 2 days after angioplasty, immediately before discharge, and again 1 month after discharge with adjustments to antiplatelet dosing as indicated. Follow-up was performed by neurologic evaluation and Doppler sonography examination at 1, 3, and 12 months after the procedure. At 5–6 months, the P2Y12 inhibitor was switched to 100 mg/d aspirin, which was maintained indefinitely.

## 3. Results

Out of 283 patients who underwent CAS during the recruitment period, 32 patients (11%, 26 males and 6 females) with carotid stenosis and nonvalvular AF who were under anticoagulant therapy satisfied inclusion and exclusion criteria. No patient in this series required bilateral carotid angioplasty. The mean age at presentation was 71 years (range 62–86 years).

The majority had known risk factors for stroke, including hypertension (28, 87.5%), hyperlipidemia (22, 68.8%), diabetes (17, 53.1%), coronary artery disease in (19, 59.3%), and a history of smoking (11, 34.3%). The CHA2DS2 VASc score was ≥4 in all patients: 4 in six patients, 5 in five, 6 in four, 7 in ten, 8 in four and 9 in three patients. All 32 patients were under DOAC treatment: 22 were treated with apixaban (Eliquis, Bristol-Myers Squibb), eight with dabigatran (Pradaxa, Boehringer Ingelheim), and two with rivaroxaban (Xarelto, Bayer). There were 21 patients (65.6%) receiving clopidogrel + aspirin, seven (21.8%) on prasugrel + aspirin, and four (12.5%) receiving ticagrelor + aspirin.

### 3.1. Clinical and Imaging Presentation

There were atherosclerotic carotid stenoses in all 32 patients, including three with post-endarterectomy restenosis; 19 (59.4%) had high-grade stenosis and 13 (40.6%) with moderate stenosis on initial angiography per NASCET criteria.

Overall, 19/32 patients (59.4%) presented with minor stroke (*n* = 8) or TIA (*n* = 11) homolateral to a carotid artery with stenosis of more than >50% (mean 78%, range 60–95%). The mean NIHSS score at presentation was 3 (range 1–5) in seven patients with ischemic stroke who could be assessed; there was insufficient data for NIHSS assessment in one patient. The interval between stroke/TIA and stenting ranged from 5 to 19 days (mean 12 days).

Among the 13 patients (30.6%) with asymptomatic carotid stenoses, seven presented severe stenosis, one with moderate-to-severe carotid stenosis (mean stenosis 78.0 ± 9.7%, range 65–95%) and contralateral carotid occlusion, and five patients who were being followed sonographically were advised to undergo stenting after carotid stenosis progression.

### 3.2. Endovascular Intervention

CAS was performed successfully in all stenosed carotid arteries. All 32 CAS procedures were performed under local anesthesia, with or without mild sedation. Carotid Wallstents (Schneider Boston Scientific, Galway, Ireland) were used in 30 arteries (91%) and were considered the first choice for symptomatic and asymptomatic patients where there was suspicion of high embolic potential. Precise stents (Cordis, Johnson & Johnson, Miami, FL, USA) were used in two cases (9%). In 30 procedures, protected pre-angioplasty proceeded stent deployment, and in two procedures (9%), pre-angioplasty preceded filter placement. In every case, stent deployment was done under filter protection. Post-angioplasty was performed at least once in 12 patients (37.5%).

Mean stenosis according to NASCET criteria was reduced from 78.0% ± 9.7% prior to treatment to 17.3% ± 12.2% after stent angioplasty.

No patient in this series complicated with myocardial infarction or embolic stroke, or developed hyperperfusion syndrome, intracranial hemorrhage, or acute/subacute stent thrombosis within 30 days after stenting. Periprocedural neurological morbidity and mortality were zero. There was transient intraprocedural hemodynamic instability (bradycardia and hypotension or bradycardia alone) during 15/32 (46.9%) procedures. Prolonged instability, which occurred in four patients (12.5%), was considered the main reason for prolonged a hospital stay in our series. Post-angioplasty hemodynamic instability was managed with antihypertensive drug dose modification or discontinuation and conservative management, with uneventful stabilization in all the cases. One patient (3%) presented a retroperitoneal hematoma that was managed conservatively. One patient (3%) underwent semi-urgent coronary angiography after atypical chest pain but there was no need for coronary intervention. All patients were discharged under double antithrombotic therapy without aspirin, DOAC, and single non-aspirin antiplatelet medication at the minimal dose required to reach and maintain a PRU of less than 150.

### 3.3. Clinical and Radiological Follow-Up

During clinical and Doppler ultrasound follow-up (mean 16 months), none of the 32 patients developed new neurological complaints, and there was no- or mild stenosis of the treated arteries. None of the patients presented hemorrhagic complications. All patients continued with double antithrombotic regimens that excluded aspirin. Antiplatelet dosing was reduced in 14 patients (43.8%) based on antiplatelet inhibition testing. In 10 (47.6%) of the 21 patients receiving clopidogrel, the dose was reduced to 75 mg every 48 h (*n* = 6) or even every 72 h (*n* = 2). The doses were maintained at 5 mg per day for all seven patients receiving prasugrel and at 60 mg every 12 h for the four patients receiving ticagrelor.

At 3–4-month follow-up examinations, 28 out 32 patients were clinically and sonagraphically stable and four patients presented moderate stenosis. They were all instructed to switch their antiplatelet agent to aspirin 100 mg/d at the beginning of month 5–6, and maintain the DOAC + aspirin indefinitely. At 12-month follow-up, all patients were clinically stable and four patients presented moderate in-stent restenosis on Doppler studies. These findings were not confirmed by CTA and the patients were managed conservatively. At the close of data gathering for this study, five patients have discontinued the antithrombotic scheme for brief periods to undergo nine surgical interventions (cardiovascular in three, dental in three, appendectomy in one, cholecystectomy in one, plastic surgery in one).

## 4. Discussion

AF is an important public health problem, contributing significantly to increasing health care costs in Western nations [12]. Its prevalence is increasing in parallel with better treatment options for chronic cardiac and noncardiac disease as well as improved ability to suspect and diagnose AF. Prevalence varies with age and gender: AF is more frequent in males and is diagnosed in 10–17% of those aged 80 years or older [12]. Patients with AF have an age-adjusted risk of stroke that is five-fold higher than the general population, regardless of the type of AF. The mechanisms underlying these strokes are multiple and disputed [13]. Cardiogenic embolism is a frequent cause of stroke in these patients, but AF could also be a marker of generalized atherosclerosis causing non-cardioembolic strokes [13,14]. Since the introduction of systematic anticoagulation therapies, the absolute number of ischemic strokes has decreased dramatically; however, stroke secondary to AF is associated with a 50% increase in the risk of serious disability [15].

Strategies for stroke prevention may be influenced by the coexistence of AF and carotid artery atherosclerosis [13], which was diagnosed in 12% of men and 11% of women over 70 with AF [16]. When patients with AF under anticoagulant therapy require carotid revascularization, carotid endarterectomy is usually favored over stenting, which requires triple antithrombotic therapy; however, there is limited literature supporting this practice. Cardiac arrhythmias were a specific exclusion factor for two of the most well-known randomized controlled trials, the Asymptomatic Carotid Atherosclerosis Study (ACAS) [17], and NASCET [18]. Although several studies have documented a significantly increased risk of death or stroke after endarterectomy in patients with AF [19,20,21,22], some of the severe complications associated with endarterectomy, such as post-CEA neck hematoma, are significantly increased in patients under perioperative anticoagulation or DAT [1,2].

In this report we present our preliminary experience in 32 patients with AF who were on DOAC regimens and underwent carotid artery stenting. All the patients were successfully managed with an abbreviated perioperative scheme of triple antithrombotic therapy (DOAC plus DAT) that was converted to double antithrombotic therapy (DOAC and a single non-aspirin antiplatelet). The choice of the antiplatelet agent and dose was based on antiplatelet inhibition testing with a target PRU of less than 150.

This approach proved to be safe and effective. There were no hemorrhagic events or embolic or thrombotic stent complications during hospital stays or long-term follow-up.

Our protocol is partially based on the experience cardiovascular surgeons and interventionalists with AF patients undergoing percutaneous coronary interventions, and those with acute coronary syndrome. In these patients, although triple antithrombotic therapy can increase the risks of bleeding and stroke, and thrombosis prevention is crucial.

Recent randomized controlled trials have shown favorable outcomes for patients with AF using dual antithrombotic strategies that include DOACs and P2Y12 inhibitors [23,24,25]. Based on these results, the most recent guidelines recommend DOAC + P2Y12 inhibitors in patients with AF who undergo percutaneous coronary interventions with stenting for acute coronary syndrome [26]. Although regimens without aspirin were shown to have lower rates of hemorrhage, it remains unknown whether any particular strategy is preferable. Recently, Kuno et al. [7], compared the efficacy and safety of nine antithrombotic strategies, including combinations of vitamin K antagonist with dual antiplatelet therapy or a P2Y12 inhibitor (clopidogrel, prasugrel, and ticagrelor), and combinations of DOAC (apixaban, dabigatran, rivaroxaban, and edoxaban) with dual antiplatelet therapy or a P2Y12 inhibitor, in patients with AF undergoing coronary interventions. The primary safety outcome was bleeding and the primary efficacy outcome was major adverse cardiovascular events. Of all the combinations, they found that apixaban plus a P2Y12 inhibitor had the lowest bleeding risk and was ranked the best treatment. There were no significant differences in ischemic outcomes of major adverse cardiovascular events between various antithrombotic regimens.

There is a general consensus that dual antiplatelet therapy should be applied in carotid artery stenting procedures and other neuroendovascular interventions; however, there are no guidelines regarding the choice of platelet inhibitors, the duration of dual antiplatelet therapy, the need for platelet function testing, or an appropriate platelet function target [27]. The inherent variability between patients in the pharmacodynamic response to common antiplatelet agents such as aspirin and clopidogrel complicates optimal selection of antiplatelet agents by clinicians. Muller-Schunck [28] found a statistical relationship between nonresponsiveness to clopidogrel and adverse thromboembolic events after supraaortic trunk stenting, and resistance to clopidogrel-induced platelet inhibition has been described in up to 40% of patients [29]. Such nonresponders would seem to be responsible for the majority of thromboembolic complications following neuroendovascular procedures [28,30]. However, as is the case for cardiovascular procedures, the association between clinical outcomes and residual platelet reactivity in patients undergoing neuroendovascular procedures requiring dual antiplatelet therapy has been explored only in small, retrospective, single-center studies. Furthermore, the collective interpretation of their findings is challenging due to many confounding variables, including the neurointerventionalists’ surgical experience, nonstandardized antiplatelet regimens, variable definitions for hyper- and hyporesponse to these regimens, and even different definitions for adverse clinical events. Platelet function testing appears to identify patients with clopidogrel resistance prior to neuroendovascular procedures. When nonresponders are converted to alternative platelet inhibitors, neurological outcomes and thromboembolic complication rates may improve.

Our preliminary experience in patients with AF managed with carotid artery stenting under heparin anticoagulation and dual antiplatelet therapy who are switched to DOAC and a non-aspirin antiplatelet with a single P2Y12 inhibitor is based on many of the aforementioned concepts: testing the effects of double antiplatelet therapy during the critical time of angioplasty and stent implant followed by rapid conversion to a single P2Y12 antiplatelet regime with proven effectiveness at the time of DOAC restart. In this way we aimed to reduce the frequency of intraprocedural thromboembolic complications with the usual dual antiplatelet protection but rapidly adapt the antithrombotic plan to the more conservative DOAC plus a single P2Y12 inhibitor as proposed for patients with AF who undergo percutaneous coronary interventions. The larger diameter and increased flow though the ICA in comparison to coronary vessels may also contribute to the higher likelihood of preserved patency despite triple antithrombotic plan downgrading.

All patients in this series were treated with a Carotid Wallstent. The results of this study cannot be extrapolated to dual-layer stents, which are known to be more thrombogenic.

## 5. Conclusions

Carotid endarterectomy in patients with AF under DOAC is associated with a significant increase in surgical risk. Based on our experience, we believe the final role of carotid artery stenting in AF patients under DOAC warrants further exploration in rigorous trials.

## Data Availability

For additional information regarding data availability, please contact the corresponding author.

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
