# Peer review of "Carotid Artery Stenting in Patients with Atrial Fibrillation: Direct Oral Anticoagulants, Brief Double Antiplatelets, and Testing Strategy"

_jcm, 2021, doi:10.3390/jcm10225242_

Round 1
Reviewer 1 Report
The topic of the paper is the antohrombotic/ anticoagulant therapy in patients undergoing carotid artery stenting and have nonvalvular AF. The authors present their limited experience in 32 patients with AF who were on direct oral anticoagulation (DOAC) and underwent carotid artery stenting. All patients received perioperative triple therapy which was then converted to a dual treatment with non-ASS antithrombotic and DOAC. There were no hemorrhagic or ischemic events during hospital stay or long-term follow up.
Thanks a lot for this interesting paper. I have some minor comments/ questions:
Clopidogrel was the first choice in all patients and was switched to Prasugrel or Ticragelor in patients who were identified as non-responders.
Can you please explain when and why you decided to use prasugrel and when and why ticragelor? What factors played a role? Prasugrel is contraindicated in stroke patients, was this taken into account when you made the choice?
The majority of patients in this series were treated with a Carotid Wallstent. However, many centers use dual layer design stents; there seems to be a difference between the carotid wallstent and dual layer stents at least in the emergency setting, particularly for thrombotic complications. The authors might mention that it is still unclear whether the positive results of the study are transferable to dual layer stents. An early reduction to mono-antiplatelet aggregation may be associated with a higher number of ischemic events in dual layer stents.
Author Response
Point 1. Good point. Prasugrel is contraindicated in patients with a history of TIA or stroke because of the increased risks of bleedings. However, it presents several administration advantages compared to Ticagrelor (cost, once a day dose, half dose posology) making its use more comfortable and compliant. In our experience, judicious use of Prasugrel was not associated with hemorrhagic complications.
Point 2. Results of this study cannot be extrapolated to double-layered stents
Reviewer 2 Report
Congratulations for this presentation, and for your general aproach.
I just would comment about the abstract. I wonder if there is a need to give so many data of the study already in the abstract, since it may distract somewhat from the very interesting conclusion you want to highligth.
Alternatively, beginning with the main result in the first sentence (indicating that you present your data on anticoagulation/antiplatelet regime for CAS and that the fact of combining apixaban with P2Y12 inhibitors in the first months after CAS has shown no complications in AF patients) would be a good option.
Author Response
We have moved details regarding patient outcome to the very early sentences of the abstract, as per the reviewer's suggestion. These statements are shown in a red font in the uploaded file.
Reviewer 3 Report
The authors report their experience in 32 patients with carotid artery stenosis treated with stenting while receiving short-course periprocedural triple antithrombotic therapy. They define the article as “retrospective study” several times throughout the manuscript; however, the report is limited to descriptive data and should be defined as “case series”.
In general, the paper is interesting but too long and sometimes repetitiv (e.g. introduction and discussion section) in the current version. I would suggest to the authors to short it removing irrelevant information in order to keep the attention of the readers.
I would also recommend english review (both language and style)
Author Response
Please see the Manuscript